

# Edge and texture aware image denoising using median noise residue U-net with hand-crafted features

Soniya S. and Sriharipriya K. C.

School of Electronics Engineering, Vellore Institute of Technology, Vellore, Tamil Nadu, India

## ABSTRACT

Image denoising is a complex task that always yields an approximated version of the clean image. Unfortunately, the existing works have focussed only on the peak signal to noise ratio (PSNR) metric and have shown no attention to edge features in a reconstructed image. Although fully convolution neural networks (CNN) are capable of removing the noise using kernel filters and automatic extraction of features, it has failed to reconstruct the images for higher values of noise standard deviation. Additionally, deep learning models require a huge database to learn better from the inputs. This, in turn, increases the computational complexity and memory requirement. Therefore, we propose the Median Noise Residue U-Net (MNRU-Net) with a limited training database without involving image augmentation. In the proposed work, the learning capability of the traditional U-Net model was increased by adding hand-crafted features in the input layers of the U-Net. Further, an approximate version of the noise estimated from the median filter and the gradient information of the image were used to improve the performance of U-Net. Later, the performance of MNRU-Net was evaluated based on PSNR, structural similarity, and figure of merit for different noise standard deviations of 15, 25, and 50 respectively. It is witnessed that the results gained from the suggested work are better than the results yielded by complex denoising models such as the robust deformed denoising CNN (RDDCNN). This work emphasizes that the skip connections along with the hand-crafted features could improve the performance at higher noise levels by using this simple architecture. In addition, the model was found to be less expensive, with low computational complexity.

## INTRODUCTION

The technique of image denoising involves taking out noise from a source image, which is essential for enhancing image quality in a variety of applications. As y = x + v, where v is additive white Gaussian noise (AWGN) with a standard deviation of sigma, the objective is to recover a clean image (x) from a noisy observation (y). Numerous strategies have been devised to tackle this issue. *Burger, Schuler & Harmeling (2012)* have shown that a multilayer perceptron (MLP) can directly learn the mapping from noisy image patches to clean patches. Another machine learning-based strategy, called the fast and flexible denoising convolutional neural network (FFDNet), is a quick and flexible denoising network that can handle different kinds and intensities of noise because it takes a tunable

Corresponding author
Sriharipriya K. C.,
sriharipriya.kc@vit.ac.in

noise level map (*Zhang, Zuo & Zhang, 2018*). Image-Prior-Based Approaches: These techniques extract meaningful information from noise in images by drawing on prior knowledge about the image, which is typically based on the Bayesian framework. To minimize an energy function that represents the properties of the image, for instance, optimization techniques are used in image denoising with Markov random fields (MRF) or conditional random fields (CRF) (*Barbu, 2009*). In addition, Total Variation (TV) regularisation techniques were presented. These techniques aim to reduce noise while maintaining significant image characteristics, such as edges. These regularisation issues have been effectively resolved by quick gradient-based algorithms (*Beck & Teboulle, 2009*).

Eventually, a method that combined traditional and learning-based approaches was created. Novel approaches have been achieved by fusing deep learning models with conventional techniques. To better model the block matching and aggregation processes, for instance, BM3D-Net expands the BM3D algorithm with a convolutional neural network (CNN) structure (*Yang & Sun, 2018*). To improve image recovery from low-resolution and noisy inputs, a deep CNN intended for denoising or super-resolution can be cascaded with an enhancement CNN (*Huang et al., 2017*).

Further emphasized multiresolution transforms. For spatially localized details like edges and singularities, these transforms offer good sparsity, which is essential for efficient denoising. By utilizing these multiresolution features, BM3D has demonstrated efficacy in identifying and conserving significant details from natural images (*Dabov et al., 2007*). Innovations in image denoising keep progressing by combining these various methods, striking a balance between noise reduction and the retention of important image elements.

The development of technology is the primary focus of researchers. One kind of deep learning model that works especially well with processing grid-like data, like images, is the deep convolutional neural network (CNN). The ability of CNNs to automatically learn spatial hierarchies of features from raw image data has revolutionized the field of computer vision. This ability makes CNNs highly effective for a variety of tasks, including object detection, segmentation, denoising, and image classification. Deep CNNs have demonstrated their efficacy in numerous computer vision applications under their capacity to represent intricate patterns and correlations found in image data. They are now a mainstay of contemporary deep learning research and development due to their adaptability and effectiveness.

Deep neural network training can improve performance, but it's a common misperception that adding more layers will automatically result in greater accuracy. As a matter of fact, the vanishing gradient problem and the curse of dimensionality problems where accuracy increases initially but rapidly decreases as network depth increases can be brought on by adding more layers (*Aadhitya & Sriharipriya, 2023*). Furthermore, prior deep learning techniques for image denoising frequently experience overfitting, which results in the loss of important edge and texture information (*Xu et al., 2023*).

This study presents a novel deep convolutional neural network (CNN) model for image denoising, called Median Noise Residue U-Net (MNRU-Net), in order to address these issues. MNRU-Net is made with a simpler architecture than traditional CNN-based denoising techniques in order to prevent training issues and maintain edge and texture

information. The model is less complicated and more computationally efficient because it makes use of a small training dataset without the need for data augmentation. Additionally, adding manually created features to the input layers of the classic U-Net model improves its learning capacity and allows for improved noise reduction and preservation of image detail.

The main contributions of our work are below

- The model assists in distinguishing between noise and significant image details by using noise estimates obtained from a median filter and image gradient information. This method enhances denoising performance, particularly at higher noise levels.
- Hand-crafted features are integrated directly into the input layers of MNRU-Net, which improves upon the conventional U-Net architecture. Feature loss in deep networks is a common problem that this integration helps to solve by improving the model's ability to preserve edges and textures.
- Without compromising denoising quality, the model keeps a straightforward architecture with fewer parameters, which makes it computationally efficient and appropriate for applications with limited resources.

## RELATED STUDY

### Convolution neural network for image processing

Due to a strong capacity to learn, CNNs have been formed for image denoising. The original distribution of noise in corrupted images may vary as a result of convolution operation, which could make training for image denoising more challenging. The batch normalization technique can address the problem by normalizing the input layers of a network. This article focuses on the internal covariant shift problem but the vanishing or exploding gradient may be severe (*Ioffe, 2015*). For retrieving smooth edges, the constrained total variation (TV) approach requires Fast gradient-based algorithms that can handle edges and eliminate noise in a corrupted image but may cause optimization problems (*Beck & Teboulle, 2009*).

To address additional inverse problems, denoiser prior can be promoted into model-based optimization techniques as a modular component (*Zhang et al., 2017b*). Alternatively, the introduction of BRDNet architecture, which integrates two networks to expand the width, addresses the mini-batch and internal covariant shift problems, reduces the training challenges of the structure, and improves performance. However, the main disadvantage of the model is more complex (*Tian, Xu & Zuo, 2020*). A dual denoising network (DudeNet) was introduced to restore clean images. A dual network with a sparse mechanism can bring out additional characteristics to improve the relevancy of the denoiser. The combination of local and global characteristics can take out salient features to restore clear details to complicated noisy images (*Tian et al., 2020b*). A framework for training DDM on a single image named SinDDM and learning the internal data of the training image utilizes a multi-scale diffusion procedure. They employ a fully convolutional lightweight denoiser, conditioned at both noise level and scale, to encourage

backscatter. With this architecture, patterns of any size, from coarse to fine, can be created (*Kulikov et al., 2023*). The use of the medical field has seen many advancements. A wide range of image-denoising methods designed with medical imaging in mind are covered in this thorough review. The effectiveness of a wide range of contemporary and conventional techniques, such as wavelet transforms, non-local means, and deep learning approaches, is assessed in various medical imaging modalities, such as magnetic resonance imaging (MRI), computed tomography (CT), and ultrasound (*Kaur & Dong, 2023*). *Dong, Ma & Basu (2021)* present a novel feature-guided CNN that is intended for denoising ultrasound images that are captured from portable devices. Because of their portable nature and their noisy operation, ultrasound machines frequently yield images of inferior quality. The feature-guided CNN improves image quality while maintaining clinically significant details by utilizing domain-specific features like edges and textures (*Dong, Ma & Basu, 2021*). *Dong & Basu (2023)* describe a denoising method for medical images that uses an explainable AI (XAI) framework to maintain significant features while denoising. To guarantee that the final images maintain crucial diagnostic features while lowering noise, the authors suggest a novel loss function that combines feature preservation with conventional denoising goals.

## Deep CNN for image denoising

A lot of research has been published using different methods of CNN, *Tian et al. (2022)* proposed that multi stage image denoising CNN with wavelet transform (MWDCNN) is a technique that consists of three stages: dynamic convolutional block (DCB), wavelet transform and enhancement block (WEB)s, and Residual block (RB) to improve the performance of denoiser. To overcome the width and depth limitations of lightweight CNNs and achieve better denoising performance, a dynamic convolution is used in the CNN and does not focus on complex validation (*Tian et al., 2022*). A new method of self-supervised image noise canceling that integrates a multi-masking approach with BSN (MM-BSN). It can be employed to resolve the issues of high noise cancellation, which other BSN methods cannot solve effectively (*Zhang et al., 2023b*). Res-WCAE with the Kullback-Leibler divergence (KLD) rule has been proposed for a lightweight and powerful deep learning architecture, particularly designed for denoising the fingerprint image. Res-WCAE contains two encoders and a decoder. It requires a large database to identify the correct fingerprint (*Liang & Liang, 2023*). Further, a data enhancement technique called recorrupted-to-recorrupted (R2R) has been proposed, to solve the problem of overfitting due to a lack of realism (*Pang et al., 2021*). *Neshatavar et al. (2022)* proposed a cyclic multi-variate function-supervised image denoising framework. The CVF-SID method can remove clear and noisy image maps from the input by exploiting several self-supervised loss metrics. For better denoising performance, deep boosting denoising net (DBDnet) has been developed. A noisy observation, creates a noise map from a residual learning network. To be more precise, it creates a raw noise map using a straightforward design, and then gradually upgrades the noise map using an enhancement function but the noise prediction may not be clear (*Ma et al., 2022*; *Quan et al., 2021*). The authors look into the possibility of complex validating CNNs for image denoise. To fully utilize the benefits of

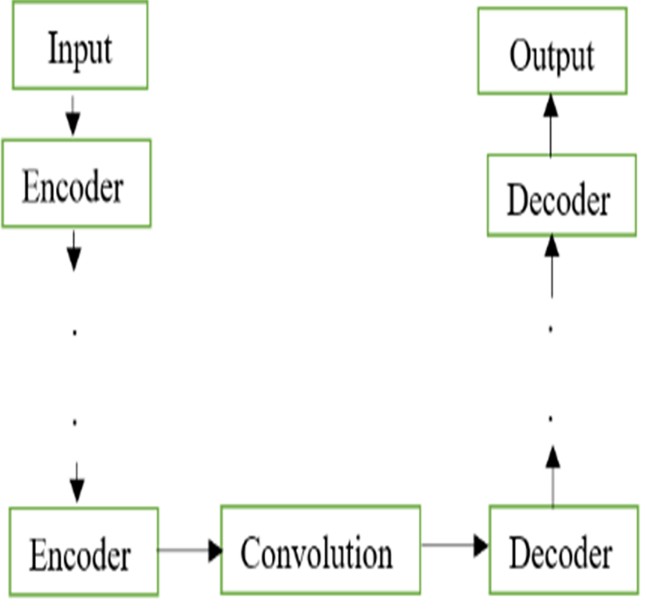

**Figure 1 Basic U-Net.** Basic U-Net architecture.

complex-valued operations, CNN was designed with its main operations defined in the complex number domain. The fact that the various models employed in this comparison study were not rigorously trained in identical settings is one of its limitations.

Moreover, the above-discussed models are increasing the performance by upgrading the networks at different levels. But it leads to complexity and more time consumption. CNNs are the most useful apparatus for denoising an image, as demonstrated by the methods-based CNNs discussed above. As a consequence, we follow suit and use a CNN to solve the image-denoising problem by constructing a simple architecture.

## MATERIALS AND METHODS

The proposed denoising model of MNRU-Net architecture has been trained and implemented. This section discussed the Basic U-Net and proposed MNRU-Net model in detail.

### Basic U-Net architecture

The U-Net models are widely used for segmentation and denoising applications. The traditional model of U-Net architecture is shown in Fig. 1. The basic architecture of the U-Net model contains an encoder and decoder constructed only using the fully convolutional layers without using any dense layers. The encoders are equipped with convolution and downsampling done by kernel filters and pooling layers respectively. In contrast to encoders, decoders are constructed using up samplers and concatenation of the features from the encoder stage. Unlike the classification models, the final output is a denoised image instead of a class label.

The major strength of deep learning models relies on their ability to learn the feature automatically without human intervention. Though the conventional U-Net models could

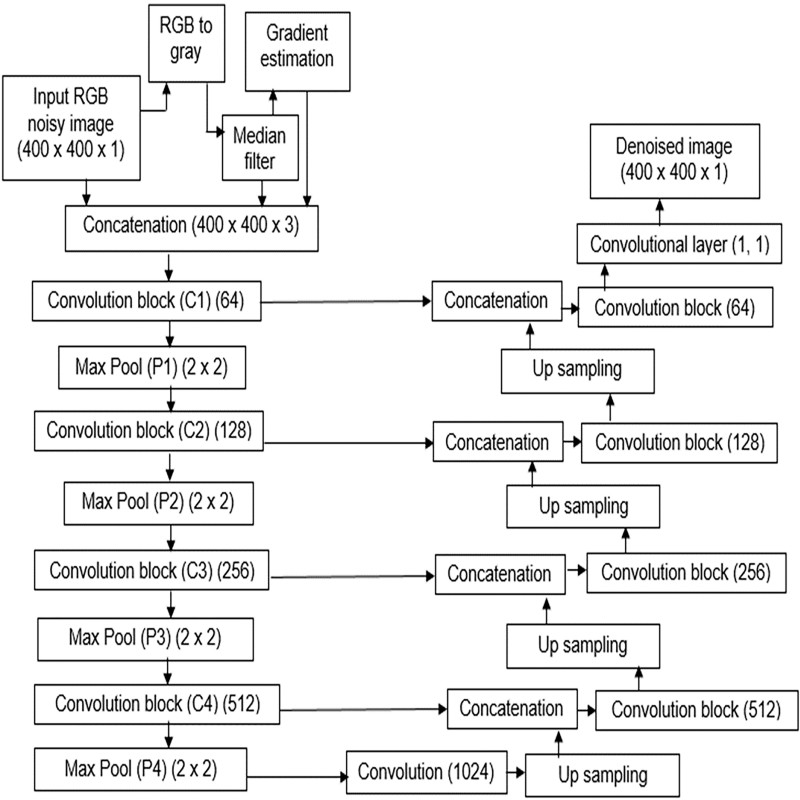

**Figure 2 Architecture of MNRU-Net.** Proposed method of MNRU-NET Design.

perform better for low-complex databases, the performance of U-Net is poor in the case of challenging applications such as denoising. Therefore, it was proposed to add hand-crafted features to the U-Net model. The U-Net model was constructed using the traditional encoders in the feature abstraction stage and the decoders were employed in the feature reconstruction stage.

## MNRU-Net architecture

The MNRU-Net model was constructed using the traditional encoders in the feature abstraction stage and the decoders were employed in the feature reconstruction stage. The traditional model of U-Net architecture as shown in Fig. 1 was modified to the MNRU-Net model as depicted in Fig. 2. The main novelty in this work was the inclusion of hand-crafted features in the deep learning model.

The idea of this research work is to supply manual features to enhance the learning capability of the conventional U-Net model. Though improved versions of U-Net models are available, the main drawback of those models is the high computation cost. In addition, the noise pattern learned by the traditional U-net model is insufficient to reconstruct a better-denoised image.

Hence, the noise pattern obtained from the conventional median filter was applied in addition to the noisy gray image. Additionally, the gradient information of the image is

also applied to the model. Hence, two image priors are evaluated and concatenated to the input gray image.

Let $I(x, y)$ be the clean image and $N(0, \sigma)$ be the noise. Then the noisy image I is represented as shown in Eq. (1).

$$I_n(x, y) = I(x, y) + N(0, \sigma). \tag{1}$$

The denoised image $I_d$ from the median filter is obtained as Eq. (2). The value of the denoised image $I_d$ at pixel location $(x, y)$ is indicated by $I_d(x, y)$. It is the result of applying the median filter at this particular pixel. The noisy image $I_n$ at pixel location $(x - k, y - l)$ is represented by the expression $I_n(x - k, y - l)$. The median is computed over a local neighborhood centered around the pixel $(x, y)$, as indicated by the coordinates $(x - k, y - l)$.

$$I_d(x, y) = Median(I_n(x - k, y - l)) \tag{2}$$

where the median represents the operation that computes the median value, and $(x - k, y - l)$ iterates over the pixels in the filter window around the pixel $(x, y)$. A $3 \times 3$ size window is used to filter the Gaussian noise. An approximate version of the noise matrix is obtained as shown in Eq. (3).

$$Noise(x, y) = I_n(x, y) - I_d(x, y). \tag{3}$$

Let $I_d(x, y)$ be the recovered image from the median filter and gradient magnitude image $G(x, y)$. The gradient magnitude is obtained as given in Eq. (4).

$$G(x, y) = \sqrt{\left(G_x(x, y)^2 + G_y(x, y)^2\right)} \tag{4}$$

where vertical gradient $G_y(x, y)$ and Horizontal gradient $G_x(x, y)$ are calculated using Sobel operators as follows:

$$
\begin{aligned}
G_x(x, y) = &(I_d(x + 1, y - 1) + 2 I_d(x + 1, y) + I_d(x + 1, y + 1)) \\
&- (I_d(x - 1, y - 1) + 2 I_d(x - 1, y) + I_d(x - 1, y + 1)
\end{aligned} \tag{4a}
$$

$$
\begin{aligned}
G_y(x, y) = &(I_d(x - 1, y - 1) + 2 I_d(x, y - 1) + I I_d(x + 1, y - 1)) \\
&- (I_d(x - 1, y + 1) + 2 I_d(x, y + 1) + I_d(x + 1, y + 1))
\end{aligned} \tag{4b}
$$

By substituting Eqs. (4a) and (4b) in Eq. (4) we get a gradient image. Hence, the MNRU-Net model is applied with three layers of information namely, the gray image, the predicted noise image from Eq. (3), and the gradient image from Eq. (4).

The structure of the encoder used in this research is shown in Fig. 3. All the convolutional layers were activated by the activation function of the rectified linear unit (ReLU) activation function. However, the final convolution layers were activated by using the 'sigmoid' function w hich yielded the final denoised image. Drop-out layers were also added in convolutional layers of both decoder and encoder to improve the performance of denoising. A dropout of 0.05 was done in the decoder and encoder stages. The encoder and

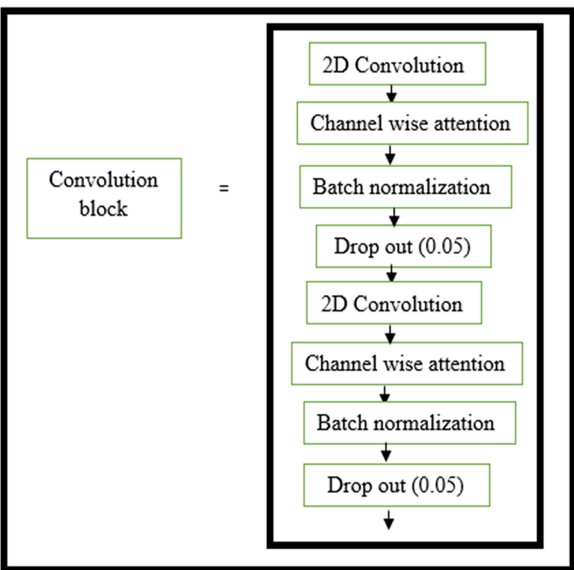

**Figure 3** **Convolution layer in encoder block.** Layers within the convolution block in the encoder block.

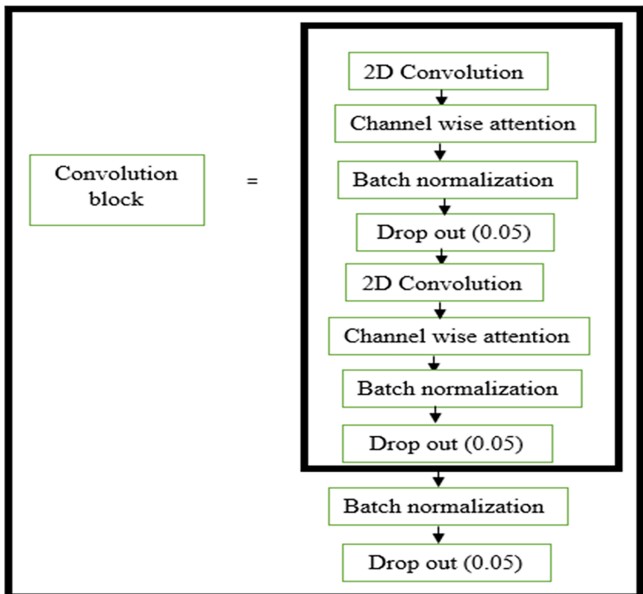

**Figure 4** **Convolution layer in decoder block.** Layers within the convolution block in the decoder block.

decoder sections are constructed using the convolution blocks C1, C2, C3, and C4 as shown in Figs. 3 and 4, respectively. Each of the convolution blocks is internally consisting of two convolutional layers. The number of filters in each encoder block is 64, 128, 256, and 512 respectively for blocks 1, 2, 3, and 4, respectively, in downstream.

Traditional deep networks may experience gradients that explode or vanish as well as becoming stuck in weak local minima when the learning rate is too high. These problems

are aided by batch normalization. Normalizing network activations prevents slight differences in layer parameters from amplifying as the data spreads throughout a deep network (*Ioffe, 2015*).

## Handcrafted features

When applying hand-crafted features to image-denoising tasks, neural networks perform much better, especially when there is a lack of training data or a high noise level. By using pre-defined features that capture significant image properties, the model can make use of handcrafted features, which minimizes the need for large datasets and intensive augmentation. For situations where data availability is restricted, MNRU-Net becomes especially helpful. In contrast to more intricate and sophisticated models, MNRU-Net achieves competitive outcomes with a more straightforward architecture. Handcrafted features reduce the need for deeper layers, making the model faster to train and less expensive to compute.

As demonstrated by the improved performance metrics (PSNR, SSIM, figure of merit) across various noise standard deviations, U-Net's architecture combined with hand-crafted features allows for better handling of high noise levels.

## Loss function

Since the problem is related to the reconstruction of the noiseless image, the loss function was chosen as mean squared error (MSE). The loss function as shown in Eq. (5) was used during the training session.

$$MSE\ loss = \frac{1}{M \times N} \sum_{i=1}^{M} \sum_{j=1}^{N} \left( predicted_{i,j} - actual\ target_{i,j} \right)^2 \tag{5}$$

where N = No of Columns, M = No of Rows, and i, j are pixel locations, $predicted_j$ = denoised images during training session, actual $target_{i,j}$ = the Original Noise free image. The main metrics were to reduce MSE loss and obtain the least error.

## Metrics evaluation

The learning rate of the model was initially set at $1 \times 10^{-3}$. This rate was adaptively modified based on the validation performance. The learning rate was reduced in non-linear steps of $1 \times 10^{-5}$, $1 \times 10^{-6}$ and the least of $1 \times 10^{-7}$. L2 regularization was set to 0.000001. This adaptive variation was done using the call-back functions when there was no improvement even after a patience of 10. The model was optimized by utilizing the Adam optimizer (*Kingma & Ba, 2014*). The batch size was adjusted to four in the case of the BSD300 database. This model was trained for 50 epochs. In the training loop, the sessions were iterated through four images per batch to cover all the training images. Since the main aim is to improve the learning capability of the traditional U-Net model, image augmentation was not done in this work.

The training metric was chosen as the peak signal to noise ratio (PSNR) as shown in Eq. (6).

$$PSNR = 10 \log \frac{Max^2}{MSE\ loss} \text{dB} \tag{6}$$

where Max = Maximum value of the pixel in the original noise-free image.

The testing performance was obtained from the images that were not exposed to training and validation. The average performance in terms of MSE and PSNR was obtained as 15,25 and 50 dB respectively. The performance of the MNRU-Net model to denoise is evaluated based on PSNR, SSIM, and FOM. In addition to the traditional metrics, a few of the feature-based metrics were also included in the performance evaluation. The formula to calculate the metrics is presented in Eqs. (7) & (8).

$$FOM = \frac{1}{\max(|G_t|\ |D_c|)} \sum \frac{1}{1 + k.d_{G_t}^2(p)} \tag{7}$$

$$SSIM\ (x, y) = \frac{\left(2\mu_x\mu_y + c_1\right)\left(2\sigma_{xy} + c_2\right)}{\left(\mu_x^2 + \mu_y^2 + c_1\right)\left(\sigma_x^2 + \sigma_y^2 + c_2\right)}. \tag{8}$$

### Training and testing datasets

The images were obtained from the Berkeley Segmentation Dataset (BSD). The images were of dimensions 481 × 321 and 321 × 481, with three color channels R, G, and B. The gray format of images was used after resizing those to 400 × 400, with an option of nearest interpolation. The database of 300 images was split into 80% of training images which accounted for 240 images. The remaining images of 30 and 30 were used for validation and testing respectively. The split of the database was done after shuffling the names of the images with a random state of 42. All the images were initially rescaled to in the range [0, 1] from its original scale of [0, 255].

For testing purposes, we used a set of four benchmark datasets they are Set12, KODAK24, McMaster, and CBSD68. The description of each dataset is given below.

**Set12:** A compact collection of 12 grayscale pictures that are frequently used to assess denoising algorithms, enabling a direct comparison with techniques that are currently in use in the literature.

**Kodak24:** This collection of 24 finely detailed color images is perfect for evaluating how well denoising techniques maintain color fidelity and sharpness.

**McMaster:** 18 color images with an emphasis on textures and high-frequency details, which is essential for determining how well fine-grained image details are preserved when denoising.

**CBSD68** is a subset of the Berkeley Segmentation Dataset made up of 68 color images that were chosen especially for testing denoising algorithms on a variety of real-world images with different content.

### RESULTS

The standard values of SSIM and PSNR of the proposed method are listed in Table 1 with different noise values. In this article, we aim to concentrate on image denoising and also,

**Table 1 Metrics evaluation.** Essential metrics of different noise levels.

| Metrics | PSNR (dB) | SSIM | FOM |
|---|---|---|---|
| σ = 15 | 30.6165 | 0.9290 | 0.9024 |
| σ = 25 | 28.6515 | 0.8960 | 0.8768 |
| σ = 50 | 26.4619 | 0.8580 | 0.8375 |

**Table 2 Comparision table for performance.** Results of average value of PSNR in (dB) with various models for noise levels of 15, 25, and 50.

| Models | BM3D (*Dabov et al., 2007*) | WNNM (*Gu et al., 2014*) | EPLL (*Zoran & Weiss, 2011*) | TNRD (*Chen & Pock, 2015*) | CSF (*Schmidt & Roth, 2014*) | MLP (*Burger, Schuler & Harmeling, 2012*) |
|---|---|---|---|---|---|---|
| σ = 15 | 31.07 | 31.37 | 31.21 | 31.42 | 31.24 | – |
| σ = 25 | 28.57 | 28.83 | 28.68 | 28.92 | 28.74 | 28.74 |
| σ = 50 | 25.62 | 25.87 | 25.67 | 25.97 | – | 26.03 |

| DnCNN (*Zhang et al., 2017a*) | IRCNN (*Zhang et al., 2017b*) | ECNDNet (*Tian et al., 2020c*) | FFDNet (*Zhang, Zuo & Zhang, 2018*) | ADNet (*Tian et al., 2020b*) | RDDCNN (*Zhang et al., 2023b*) | MNRUNet |
|---|---|---|---|---|---|---|
| 31.72 | 31.63 | 31.71 | 31.63 | 31.74 | 31.76 | 30.61 |
| 29.23 | 29.15 | 29.22 | 29.19 | 29.25 | 29.27 | 28.65 |
| 26.23 | 26.19 | 26.23 | 26.29 | 26.29 | 26.30 | 26.94 |

we are preserving the edge and texture information in detail. Experimental results also show a denoising image with clear edges. Most of the traditional methods did not calculate FOM. However, this current work focuses on evaluating SSIM and FOM along with the traditional PSNR metric.

## DISCUSSION

In this research work, the suggested method is correlated with recent state-of-the-art works in image-denoising techniques such as *robust deformed denoising CNN* (RDDCNN; *Zhang et al., 2023a*), attention-guided denoising convolutional neural network (ADNet; *Tian et al., 2020a*), enhanced convolutional neural denoising network (ECNDNet; *Tian et al., 2020c*) and conventional denoising techniques such as denoising convolutional neural networks (DnCNN; *Zhang et al., 2017a*), image restoration CNN (IRCNN; *Zhang et al., 2017b*), Expected Patch Log-Likelihood (EPLL; *Zoran & Weiss, 2011*), a cascade of shrinkage fields (CSF; *Schmidt & Roth, 2014*), Trainable Nonlinear Reaction Diffusion (TNRD; *Chen & Pock, 2015*), and BM3D (*Dabov et al., 2007*) to demonstrate the effect of the proposed method. Table 2 shows several network performances with different noise values of 15, 25 and 50, respectively.

The performance of the network can be analyzed by evaluating PSNR values with different noise levels of 15, 25, and 50 respectively. Figure 5 demonstrates that our proposed work performed well compared to other traditional methods. Graphical representation clearly shows the performance of MNRU-Net when the noise level of

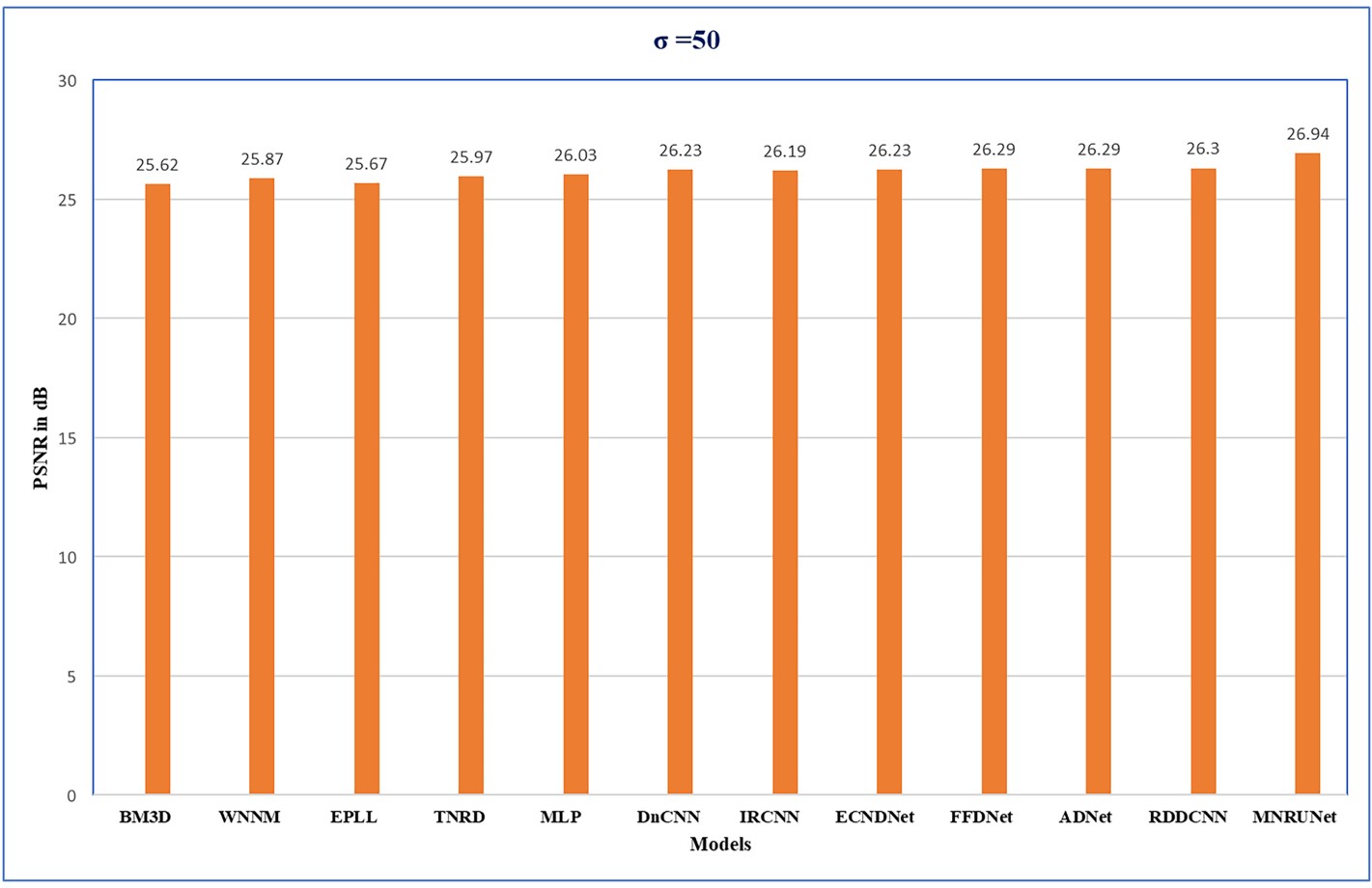

**Figure 5 Performance evaluation.** Performance analysis of several methods with Average PSNR (dB) when noise level of 50.

$\sigma$ = 50. For higher noise levels our models get better PSNR values by using simple architecture are compared to the various complex models. However, the proposed MNRU-Net with noise levels of 15 and 25 is getting nearer the PSNR values of other models.

Figure 6 shows the training and validation performance. Throughout the training process, the model continuously increases PSNR and lowers MSE, exhibiting good generalization across both noise levels. The validation curves quickly stabilized, indicating that even with different noise levels, the model is very effective at learning and reducing overfitting. The model balances training accuracy and validation performance, as evidenced by the small difference between training and validation metrics, and shows good generalization across both noise levels.

Using the Set12 dataset, we tested our work at noise levels of 50, 25, and 15. Table 3 describes each image value and displays the average PSNR value for each model. Based on the comparison table, we find that, with minor variations in 15, our MNRU-Net architecture performs better at noise levels of 50 and 25.

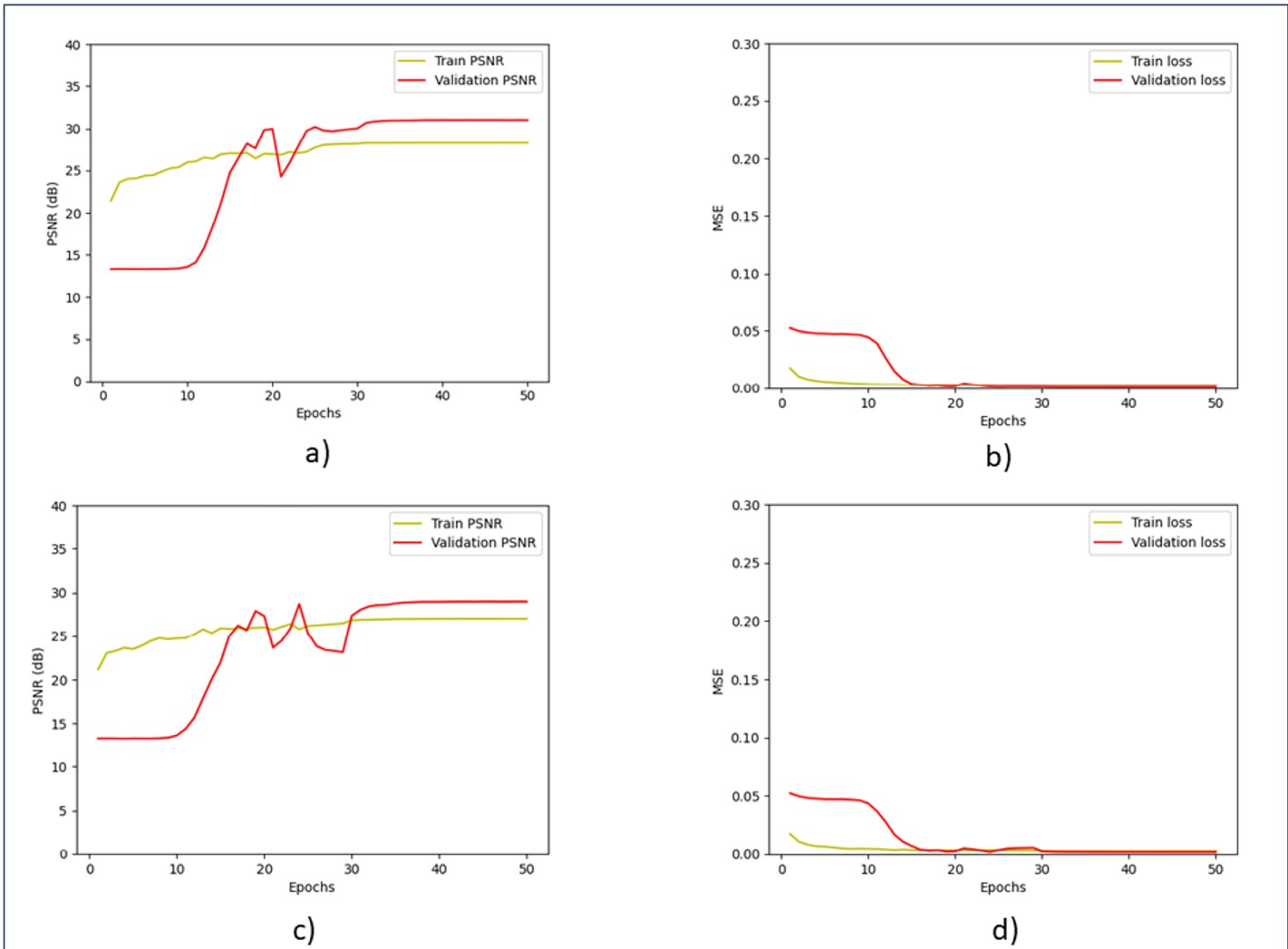

**Figure 6 Training and validation of PSNR and MSE.** (A) Training and validation of PSNR at the noise level of 15. (B) Training and validation of MSE at the noise level of 15. (C) Training and validation of PSNR at the noise level of 25. (D) Training and validation of MSE at the noise level of 25.

**Table 3 Set12 dataset performance.** Average results for PSNR (dB) of different models from the Set12 dataset when noise levels of 15, 25, and 50.

**Noise level =15**

| Images | BM3D (*Dabov et al., 2007*) | WNNM (*Gu et al., 2014*) | EPLL (*Zoran & Weiss, 2011*) | CSF (*Schmidt & Roth, 2014*) | TNRD (*Chen & Pock, 2015*) | DnCNN (*Zhang et al., 2017a*) | IRCNN (*Zhang et al., 2017b*) | FFDNet (*Zhang, Zuo & Zhang, 2018*) | ECNDNet (*Tian et al., 2020c*) | RDDCNN (*Zhang et al., 2023a*) | MNRUNet |
|---|---|---|---|---|---|---|---|---|---|---|---|
| C.man | 31.91 | 32.17 | 31.85 | 31.95 | 32.19 | 32.61 | 32.55 | 32.43 | 32.56 | 32.61 | 31.69 |
| Parrot | 31.37 | 31.62 | 31.42 | 31.37 | 31.63 | 31.83 | 31.84 | 31.81 | 31.82 | 31.93 | 31.97 |
| House | 34.93 | 35.13 | 34.17 | 34.39 | 34.53 | 34.97 | 34.89 | 35.07 | 34.97 | 35.01 | 34.59 |
| Lena | 34.26 | 34.27 | 33.92 | 34.06 | 34.24 | 34.62 | 34.53 | 34.62 | 34.52 | 34.57 | 32.57 |

(Continued)

**Noise level =15**

| Images | BM3D (Dabov et al., 2007) | WNNM (Gu et al., 2014) | EPLL (Zoran & Weiss, 2011) | CSF (Schmidt & Roth, 2014) | TNRD (Chen & Pock, 2015) | DnCNN (Zhang et al., 2017a) | IRCNN (Zhang et al., 2017b) | FFDNet (Zhang, Zuo & Zhang, 2018) | ECNDNet (Tian et al., 2020c) | RDDCNN (Zhang et al., 2023a) | MNRUNet |
|---|---|---|---|---|---|---|---|---|---|---|---|
| Peppers | 32.69 | 32.99 | 32.64 | 32.85 | 33.04 | 33.30 | 33.31 | 33.25 | 33.25 | 33.31 | 32.15 |
| Barbara | 33.10 | 33.60 | 31.38 | 31.92 | 32.13 | 32.64 | 32.43 | 32.54 | 32.41 | 32.62 | 28.53 |
| Starfish | 31.14 | 31.82 | 31.13 | 31.55 | 31.75 | 32.20 | 32.02 | 31.99 | 32.17 | 32.13 | 32.56 |
| Boat | 32.13 | 32.27 | 31.93 | 32.01 | 32.14 | 32.42 | 32.34 | 32.38 | 32.37 | 32.42 | 30.97 |
| Monarch | 31.85 | 32.71 | 32.10 | 32.33 | 32.56 | 33.09 | 32.82 | 32.66 | 33.11 | 33.13 | 32.92 |
| Man | 31.92 | 32.11 | 32.00 | 32.08 | 32.23 | 32.46 | 32.40 | 32.41 | 32.39 | 32.38 | 31.40 |
| Airplane | 31.07 | 31.39 | 31.19 | 31.33 | 31.46 | 31.70 | 31.70 | 31.57 | 31.70 | 31.67 | 32.84 |
| Couple | 32.10 | 32.17 | 31.93 | 31.98 | 32.11 | 32.47 | 32.40 | 32.46 | 32.39 | 32.46 | 30.77 |
| Average | 32.37 | 32.70 | 32.14 | 32.32 | 32.50 | 32.86 | 32.77 | 32.77 | 32.81 | 32.85 | 31.91 |
| **Noise level = 25** | | | | | | | | | | | |
| C.man | 29.45 | 29.64 | 29.26 | 29.48 | 29.72 | 30.18 | 30.08 | 30.10 | 30.11 | 30.20 | 29.93 |
| Parrot | 28.93 | 29.15 | 28.95 | 28.90 | 29.18 | 29.43 | 29.47 | 29.44 | 29.38 | 29.53 | 29.76 |
| House | 32.85 | 33.22 | 32.17 | 32.39 | 32.53 | 33.06 | 33.06 | 33.28 | 33.08 | 33.13 | 32.14 |
| Lena | 32.07 | 32.24 | 31.73 | 31.79 | 32.00 | 32.44 | 32.43 | 32.57 | 32.38 | 32.40 | 30.41 |
| Peppers | 30.16 | 30.42 | 30.17 | 30.32 | 30.57 | 30.87 | 30.88 | 30.93 | 30.85 | 30.82 | 30.03 |
| Barbara | 30.71 | 31.24 | 28.61 | 29.03 | 29.41 | 30.00 | 29.92 | 30.01 | 29.84 | 30.03 | 26.59 |
| Starfish | 28.56 | 29.03 | 28.51 | 28.80 | 29.02 | 29.41 | 29.27 | 29.32 | 29.43 | 29.38 | 29.84 |
| Boat | 29.90 | 30.03 | 29.74 | 29.76 | 29.91 | 30.21 | 30.17 | 30.25 | 30.14 | 30.19 | 28.91 |
| Monarch | 29.25 | 29.84 | 29.39 | 29.62 | 29.85 | 30.28 | 30.09 | 30.08 | 30.30 | 30.36 | 30.57 |
| Man | 29.61 | 29.76 | 29.66 | 29.71 | 29.87 | 30.10 | 30.04 | 30.11 | 30.03 | 30.05 | 29.30 |
| Airplane | 28.42 | 28.69 | 28.61 | 28.72 | 28.88 | 29.13 | 29.12 | 29.04 | 29.07 | 29.05 | 30.67 |
| Couple | 29.71 | 29.82 | 29.53 | 29.53 | 29.71 | 30.12 | 30.08 | 30.20 | 30.03 | 30.10 | 28.62 |
| Average | 29.97 | 30.26 | 29.69 | 29.84 | 30.06 | 30.43 | 30.38 | 30.44 | 30.39 | 30.44 | 29.73 |
| **Noise level = 50** | | | | | | | | | | | |
| C.man | 26.13 | 26.45 | 26.10 | 26.37 | 26.62 | 27.03 | 26.88 | 27.05 | 27.07 | 27.16 | 27.72 |
| Parrot | 25.90 | 26.14 | 25.95 | 26.12 | 26.16 | 26.48 | 26.55 | 26.57 | 26.32 | 26.53 | 27.63 |
| House | 29.69 | 30.33 | 29.12 | 29.64 | 29.48 | 30.00 | 29.96 | 30.37 | 30.12 | 30.21 | 29.63 |
| Lena | 29.05 | 29.25 | 28.68 | 29.32 | 28.93 | 29.39 | 29.40 | 29.66 | 29.29 | 29.32 | 28.36 |
| Peppers | 26.68 | 26.95 | 26.80 | 26.68 | 27.10 | 27.32 | 27.33 | 27.54 | 27.30 | 27.38 | 28.20 |
| Barbara | 27.22 | 27.79 | 24.83 | 25.24 | 25.70 | 26.22 | 26.24 | 26.45 | 26.26 | 26.36 | 25.23 |
| Starfish | 25.04 | 25.44 | 25.12 | 25.43 | 25.42 | 25.70 | 25.57 | 25.75 | 25.72 | 25.72 | 28.12 |
| Boat | 26.78 | 26.97 | 26.74 | 27.03 | 26.94 | 27.20 | 27.17 | 27.33 | 27.16 | 27.23 | 26.90 |
| Monarch | 25.82 | 26.32 | 25.94 | 26.26 | 26.31 | 26.78 | 26.61 | 26.81 | 26.82 | 26.84 | 28.62 |
| Man | 26.81 | 26.94 | 26.79 | 27.06 | 26.98 | 27.24 | 27.17 | 27.29 | 27.11 | 27.22 | 27.56 |
| Airplane | 25.10 | 25.42 | 25.31 | 25.56 | 25.59 | 25.87 | 25.89 | 25.89 | 25.79 | 25.88 | 28.04 |
| Couple | 26.46 | 26.64 | 26.30 | 26.67 | 26.50 | 26.90 | 26.88 | 27.08 | 26.84 | 26.88 | 26.79 |
| Average | 26.72 | 27.05 | 26.47 | 26.78 | 26.81 | 27.18 | 27.14 | 27.32 | 27.15 | 27.23 | 27.73 |

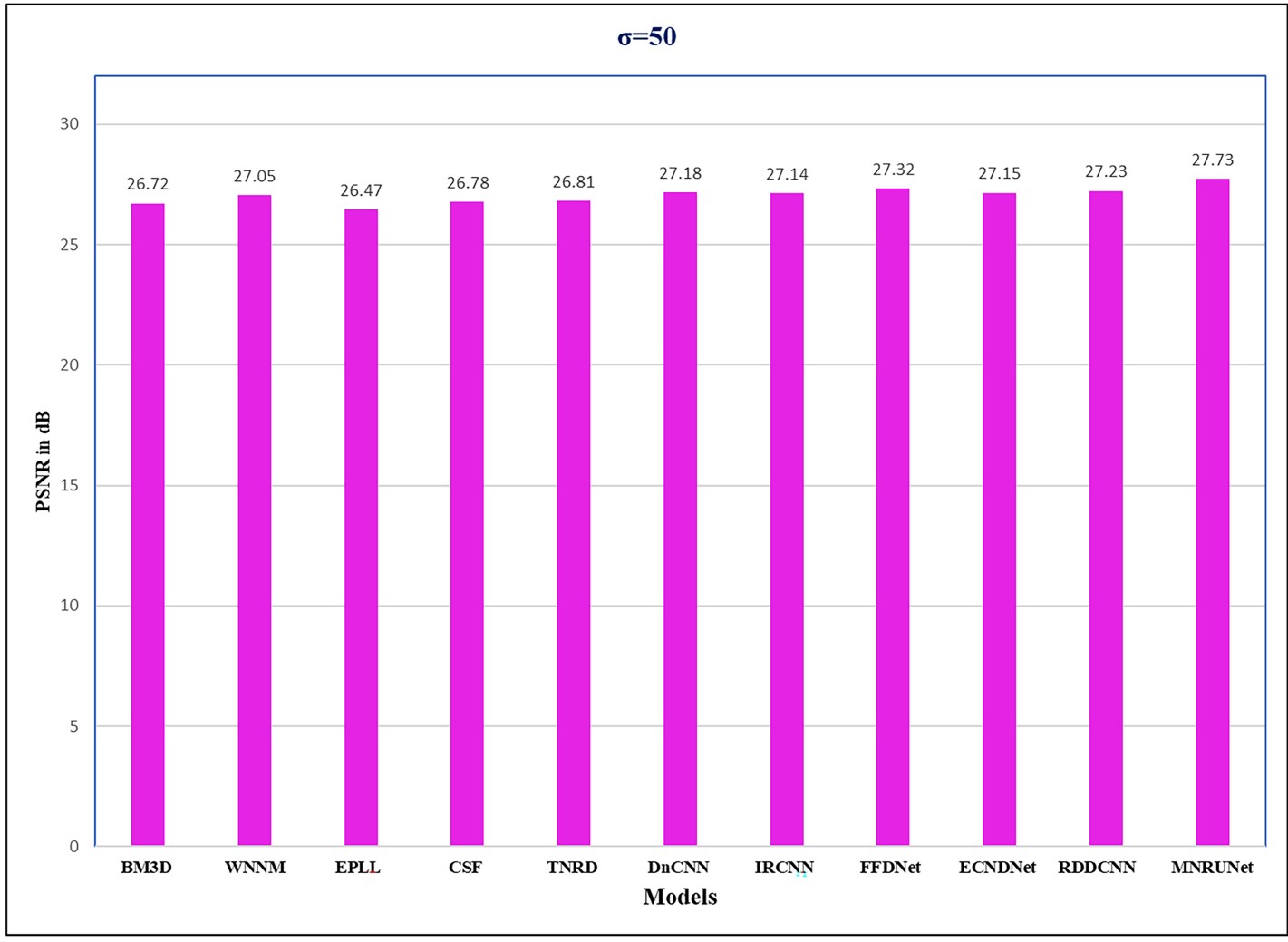

**Figure 7 Performance analysis of the Set12 dataset.** Performance analysis of several methods from the Set12 dataset with average PSNR (dB) with noise level of 50.

Our proposed model is more satisfactory for gray image denoising, as evidenced by the figures, which show that the clean images obtained through its use are clearer than those obtained through additional methods. The performance analysis has been mentioned in Fig. 7, it shows a higher performance compared to the other state-of-the-art methods when the noise level of 50 from the Set12 dataset. This performance analysis shows how our architecture works at higher noise levels. In the research work, we focussed on the proposed method as efficient, worked out with different datasets, and got better PSNR values listed in tables and expressed in figures and graphs.

Table 4 shows, that our model works well in any database with better PSNR values. Also, the model is less complex and computationally less expensive.

The proposed model is designed to achieve greater performance with simple architecture and also concentrates on edge and texture-aware information. By evaluating

**Table 4 Performance analysis using different dataset.** Results for average PSNR (dB) of MNRU-Net from the McMaster, Kodak24, and CBSD68 datasets with different noise levels of 15, 25, and 50.

| Noise levels | σ = 15 | σ = 25 | σ = 50 |
|---|---|---|---|
| CBSD68 | 30.89 | 28.92 | 27.23 |
| Kodak24 | 30.00 | 28.09 | 26.36 |
| McMaster | 30.71 | 29.14 | 27.06 |

**Table 5 Comparison of SSIM values to the state-of-the-art methods.**

| Models | EPLL (*Zoran & Weiss, 2011*) | MLP (*Burger, Schuler & Harmeling, 2012*) | DnCNN (*Zhang et al., 2017a*) | IrCNN (*Zhang et al., 2017b*) | MNRU-Net |
|---|---|---|---|---|---|
| σ = 15 | 0.8826 | 0.8727 | 0.9018 | 0.9071 | 0.9265 |
| σ = 25 | 0.8125 | 0.8432 | 0.8802 | 0.8562 | 0.8893 |
| σ = 50 | 0.6917 | 0.7312 | 0.7493 | 0.7500 | 0.7832 |

edge performance, we calculated Structural Similarity Index (SSIM) values and compared them with the state-of-the-art methods. Table 5 denotes the SSIM value of the MNRU-Net architecture.

## Advantages and disadvantages of MNRU-Net

By using handcrafted features in addition to learned features, MNRU-Net improves denoising performance, particularly in high-noise scenarios. This results in the denoised image's edges and textures being better preserved. MNRU-Net maintains a simpler architecture when contrasted with more intricate deep-learning models. Its simplicity makes it more accessible and faster to train, especially on constrained hardware, by lowering computational costs and memory requirements. In conclusion, MNRU-Net has a lot to offer in terms of improved performance and decreased complexity even with small amounts of data, but it also has drawbacks in terms of feature design, generalization, and possible redundancy with learned features.

## CONCLUSIONS

Our proposed method has proven a better performance with a simple model. We reconstructed a simple U-Net model as MNRU-Net with a limited training database without involving any image augmentation. To boost the learning capability of the network, hand-crafted features were added to the input layers. It was witnessed in this research work, that the MNRU-Net performs well. Also, we focussed on edge and texture-aware image denoising. In terms of image deblurring and image denoising, the suggested model is resistant to changes in noise level, image content, and hyperparameters. Based on experimental results, the suggested method outperforms other currently used techniques in terms of SSIM and PSNR, while also retaining more information on image and achieving greater performance in noise removal. In the future, we intend to handle more complicated real image denoising problems, like blurry and low-vision images.

### Funding
The authors received no funding for this work.

### Competing Interests
The authors declare that they have no competing interests.

### Author Contributions
- Soniya S. conceived and designed the experiments, performed the computation work, prepared figures and/or tables, authored or reviewed drafts of the article, and approved the final draft.
- Sriharipriya K. C. performed the experiments, analyzed the data, prepared figures and/or tables, authored or reviewed drafts of the article, and approved the final draft.

### Data Availability
The BSD dataset is available at: https://www2.eecs.berkeley.edu/Research/Projects/CS/vision/bsds.

For testing, we used SET12, CBSD68, KODAK24, and McMaster datasets from Kaggle. They are available at Figshare: S, Soniya; K C, SRIHARIPRIYA (2024). Datasets for Testing Denoised Image. figshare. Dataset. https://doi.org/10.6084/m9.figshare.26827765.v1.

### Supplemental Information
Supplemental information for this article can be found online at http://dx.doi.org/10.7717/peerj-cs.2449#supplemental-information.

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
