# Peer review of "Edge and texture aware image denoising using median noise residue U-net with hand-crafted features"

_PeerJ Computer Science, doi:10.7717/peerj-cs.2449_

## Round 0.1 · original submission · Major Revisions

· Academic Editor

Major Revisions

Thank you for submitting your manuscript to PeerJ Computer Science. The review process has been completed, and we have carefully considered the feedback provided by the reviewers.

The reviewers have acknowledged the potential value of your work but have raised several significant concern.

In light of these comments, I am recommending that your manuscript undergoes a major revision. We encourage you to carefully address each of the reviewers’ comments. A detailed response to the reviewers, explaining the changes made or providing justifications for any unaddressed points, should accompany your revised submission.

Once the revisions have been completed, your manuscript will undergo a further round of review to ensure that all major concerns have been satisfactorily addressed.

We appreciate the effort that you have put into this research and look forward to receiving your revised manuscript.

·

Basic reporting

Hello Authors,

Your idea of enhancing the denoising capability of existing neural networks by incorporating handcrafted features is interesting. However, to consider your manuscript for publication, several revisions are necessary. Here are my suggestions:

1. In the Introduction section, please provide a more specific introduction to your work. For instance, you could list your contributions. This will help me better understand your main idea.
2. The current layout of the article is quite disorganized, with figures not in their proper places, making it difficult for me to read your figures and tables.
3. Make some adjustments to the academic writing. For example, on line 109, "this author..." is incorrect. You should say "author A et al. proposed..." Additionally, you have used abbreviations such as DCB and WEBs; make sure to use the full name the first time they appear.
4. Since your work is related to image features in denoising, please add the following relevant papers to the Related Work section. These works use explainable AI feature detection mechanisms to enhance denoising results, which are related to your work. Please search for them on Google Scholar.

1)Feature-guided CNN for denoising images from portable ultrasound devices
2)Medical Image Denoising via Explainable AI Feature Preserving Loss
3)A complete review on image denoising techniques for medical images

5. Add a Discussion section to explain the reasons why handcrafted features can improve denoising capability, as well as the advantages and disadvantages of your method.

Experimental design

no comment

Validity of the findings

no comment

Additional comments

no comment

Reviewer 2 ·

Basic reporting

This paper presents an MNRU-Net model combined with handcrafted features (namely gradient image and denoised image using a Median filter of size 3 by 3) to denoise images. Here are my overall comments:
1. There are some font inconsistencies between the main text and citation; for example, different font types or different sizes are being used throughout the paper.
2. Some of the abbreviations are never defined. For example, what does RDDCNN stand for?
3. There are lots of unnecessary and inconsistent symbols in Eq.1 to 4b, for example:
- In equation 2, I assume 1 is supposed to be l, also what is k and l represent?
-There is In in Eq.1 and I_noisy in Eq.2.
- Similarly I_d in Eq. 2 and I_denoised in Eq.3

Experimental design

1. The comparison with other methods only shows PSNR metrics. Other metrics such as SSIM should be included for a better understanding and comparison.
2. More details about the dataset should be given in the paper.

Validity of the findings

1. Why are you using the median filter to obtain the denoised image and not Gaussian smoothing? It is well known that the median filter works best for impulse noise.
2. Why did you stick with 3 by 3 median filters? You could have used an adaptive size that would work better for noisier images.
9. An argument about why the proposed model is performing better in terms of PSNR compared to the other methods when the noise level is higher is needed.

---

## Round 0.2 · accepted · Accept

· Academic Editor

Accept

I hope this message finds you well. After carefully reviewing the revisions you have made in response to the reviewers' comments, I am pleased to inform you that your manuscript has been accepted for publication in PeerJ Computer Science.

Your efforts to address the reviewers’ suggestions have significantly improved the quality and clarity of the manuscript. The changes you implemented have successfully resolved the concerns raised, and the content now meets the high standards of the journal.

Thank you for your commitment to enhancing the paper. I look forward to seeing the final published version.

·

Basic reporting

Thank you very much to the authors for their response. The concerns I raised have been mostly addressed by the authors. I agree that this manuscript now meets the standards for publication in PeerJ. I recommend this paper for publication. I also hope the authors will continue to contribute to the field of image denoising research in the future.

Experimental design

Experimental design is reasonable.

Validity of the findings

Contribution is relatively good.

Reviewer 2 ·

Basic reporting

The authors have improved their manuscript significantly.

Experimental design

The experimental design is improved and the authors included more experiments to consider other evaluation metrics.

Validity of the findings

no comment

Additional comments

no comment